# Mindfulness supports emotional resilience in children during the COVID-19 pandemic

Isaac N. Treves[1,2,3]*, Cindy E. Li[3], Kimberly L. Wang[1,2], Ola Ozernov-Palchik[1,2], Halie A. Olson[1,2], John D. E. Gabrieli[1,2]

1 Department of Brain and Cognitive Sciences, Massachusetts Institute of Technology, Cambridge, Massachusetts, United States of America, 2 McGovern Institute for Brain Research, Massachusetts Institute of Technology, Cambridge, Massachusetts, United States of America, 3 Hock E. Tan and K. Lisa Yang Center for Autism Research, Massachusetts Institute of Technology, Cambridge, Massachusetts, United States of America

* treves@mit.edu

## Abstract

An important aspect of mental health in children is emotional resilience: the capacity to adapt to, and recover from, stressors and emotional challenges. Variation in trait mindfulness, one's disposition to attend to experiences with an open and nonjudgmental attitude, may be an important individual difference in children that supports emotional resilience. In this study, we investigated whether trait mindfulness was related to emotional resilience in response to stressful changes in education and home-life during the COVID-19 pandemic in the United States. We conducted a correlational study examining self-report data from July 2020 to February 2021, from 163 eight-to ten-year-old children living in the US. Higher trait mindfulness scores correlated with less stress, anxiety, depression, and negative affect in children, and lower ratings of COVID-19 impact on their lives. Mindfulness moderated the relationship between COVID-19 child impact and negative affect. Children scoring high on mindfulness showed no correlation between rated COVID-19 impact and negative affect, whereas those who scored low on mindfulness showed a positive correlation between child COVID-19 impact and negative affect. Higher levels of trait mindfulness may have helped children to better cope with a wide range of COVID-19 stressors. Future studies should investigate the mechanisms by which trait mindfulness supports emotional resilience in children.

## Introduction

Mindfulness has been defined as "paying attention in a particular way: on purpose, in the present moment, and non-judgmentally" (p.4) [1]. The benefits of mindfulness in children have been examined through the study of trait mindfulness and mindfulness training. Trait mindfulness refers to one's disposition or tendency to attend to experiences with an open and non-judgmental attitude [2] and can be measured by self-report questionnaires [3–5]. Mindfulness training aims to cultivate trait mindfulness through instruction and practice. Studies of both trait mindfulness and mindfulness instruction have documented benefits to child mental

**Data Availability Statement:** Data and code are publicly available on publication at https://osf.io/48bk3/.

**Funding:** This research was funded by the Chan Zuckerberg Initiative as part of the Reach Every Reader Project, National Institutes of Health (F32-HD100064 to OO) https://reacheveryreader.gse.harvard.edu/, and the National Science Foundation Graduate Research Fellowship (Grant No. 1745302 to HO), https://www.nsfgrfp.org/. The funders had no role in study design, data collection and analysis, decision to publish, or preparation of the manuscript.

**Competing interests:** The authors have declared that no competing interests exist.

health [4, 6, 7]. An important aspect of mental health in children is *emotional resilience*, the capacity to overcome emotional challenges. Greater trait mindfulness in children has been associated with more resilience in response to bullying [8]. Here we asked whether greater trait mindfulness was a source of emotional resilience in response to stressful changes in education and home-life during the COVID-19 pandemic in the United States.

The COVID-19 pandemic severely and suddenly altered the educational, social, and home experiences of children. These changes presented serious challenges to mental health, including fear of health risks, social isolation, and more. Soon after the outbreak of COVID-19 in February and March 2020, studies reported increased loneliness, anxiety, depression, and sleep disturbances in adults complying with strict social distancing protocols in China [9]. As the pandemic unfolded, increased rates of mental health symptoms were observed in other countries around the world [10]. Some findings suggested increased prevalence of psychiatric symptoms like depression in older children [11, 12] and in girls [13]. Other studies reported that some children experienced *improvements* in mental health outcomes during social distancing [14–16]. Further research is needed to understand why some children may be more resilient than others to the mental health challenges of the pandemic.

In this study, we operationalized emotional resilience as positive mental health outcomes in the context of adverse situations [17]. We did not measure resilience as a trait, but instead measured trait mindfulness as one of the cognitive and self-regulatory skills that contributes to resilience [18]. Researchers have highlighted mindfulness as a type of cognitive regulation of emotions [19], leading to a less self-critical or judgmental attitude towards one's experiences [20, 21] and greater attention to body sensations involved in emotion [22]. Mindfulness may decrease rumination, stopping repetitive cycles of depressive thoughts, e.g., "I'm worthless"; or anxious thoughts, e.g., "What if I never get it right?", "Everything is going to fall apart" [23, 24]. Most of these studies have examined adults, but there is also some evidence that greater trait mindfulness in children is associated with less rumination and healthier patterns of emotion regulation [25]. Trait mindfulness may support emotional resilience in children.

In this study, we examined children's mental health outcomes during the COVID-19 pandemic. The 163 eight-to-ten-year-old children were from families that volunteered to participate in a randomized controlled trial (RCT) examining the potential benefits of an audiobook intervention on reading outcomes. Data were collected remotely during the height of pandemic distancing restrictions, from July 2020 to February 2021. We assessed mental health using standardized self-report measures of anxiety, depression, and stress, and asked questions about negative affect. We also measured how strongly children felt they have been impacted by COVID-19. We assessed trait mindfulness with the Child and Adolescent Mindfulness Measure (CAMM), which emphasizes nonjudgmental, accepting attention to emotions [4].

We hypothesized that trait mindfulness would negatively correlate with self-report measures of anxiety, depression, stress and negative affect as has been reported before in children [4, 25]. Critically, we hypothesized that greater trait mindfulness would provide greater resilience to the changes of the COVID-19 pandemic by specifically moderating the relationship between self-reported COVID-19 impact and negative mental health outcomes.

## Materials and methods

### Participants

We analyzed the pre-intervention (baseline) data from children and their parents in a remote reading intervention study, collected from July 2020 through February 2021. Third and fourth-grade children across the United States were recruited through Facebook ads, schools, flyers, and word of mouth. We aimed to recruit children from lower socioeconomic status

(SES) backgrounds and therefore reached out to school districts with high percentages of students eligible for free/reduced lunch and targeted Facebook ads to lower income zip codes across country (for more on recruitment strategies see [26]). Although we used multiple methods to target lower income families, the median income of our obtained sample was not as low as anticipated. Interested parents filled out an eligibility screen and were invited to participate in a baseline testing session if initial inclusionary criteria were met (see S1 Text in **S1 File**).

We collected demographic information regarding participant gender, age, grade, mental health diagnosis, parent education level and annual household income. Parents reported their child's gender, and we did not inquire about sex assigned at birth. One self-report measure we used was normed by age and sex to derive T-scores–we used the reported gender of the child.

This project received Institutional Review Board approval at the Massachusetts Institute of Technology (MIT) Committee on the Use of Humans as Experimental Subjects. We obtained informed consent from the parents of all participants as well as assent from the children over Zoom before they participated in the study. Caregivers were first asked for their permission to have the Zoom session recorded. Then the test administrator obtained verbal consent from the caregiver, and verbal assent from the child, by reviewing the respective consent forms with the caregiver and child (which had been emailed to the family beforehand).

## Measures

We obtained self-reports through online questionnaires administered remotely through Research Electronic Data Capture (REDCap). REDCap is a secure, web-based application designed to support data capture for research studies [27]. Child questionnaires were administered over Zoom, a secure video chat platform. Experimenters read the assessment questions to the children who followed along on their computer screen and answered aloud. Children were compensated 20 dollars per hour of testing.

## Child COVID Impact

To assess the impact of COVID-19 on children we devised a brief Child COVID Impact Scale. This scale consists of 7 items which are rated on a 4-point Likert scale, ranging from 'not at all' to 'a lot' (see S2 Text in **S1 File**). The initial question asks how much the child has been bothered by COVID and the following questions pertain to the impact of COVID on their daily activities and their ability to see family members and friends. For each of these categories the child is first asked how much harder it has been to engage in their daily activities, or to see family or friends, and then is asked how much they are bothered by those specific changes. Children who respond to the initial question of each category with 'not at all' are not asked the follow-up question. In those cases, the follow-up questions are automatically assigned the lowest rating of '1'. Scores range from 7 to 28, with higher scores indicative of greater COVID impact. This scale demonstrated a Cronbach's alpha of .73.

## Depression & anxiety

To measure child-reported anxiety and depressive symptoms we used the 25-item Revised Child Anxiety and Depression Scale (RCADS-25-C) [28], including the following scales: Anxiety Total scale and Depression Total scale. All items are based on Diagnostic and Statistical Manual of Mental Disorders–Fourth edition criteria [29]. The items in the Anxiety Total scale measure a "broad anxiety" dimension, assessing a variety of anxiety symptoms. The items in the Depression Total scale measure symptoms of major depressive disorder. Higher scores represent a greater degree of symptoms. Raw summative scores from each of the scales are

converted into *T*-scores, with the following ranges: low severity (0–64), medium severity (65–70), and high severity (>70). *T*-scores of medium severity are considered to be "borderline clinical threshold" whereas *T*-scores of high severity are considered to be "above clinical threshold" [30]. For this study the alpha coefficients were .74 and .70 for the Anxiety and Depression scores respectively.

### Negative affect

To measure children's affect we administered a brief 13-item questionnaire [31]. Children rated the degree to which they felt each of thirteen emotions in the past week, using a 5-point Likert scale ranging from 'almost never' to 'almost always.' We calculated a Negative Affect score from the seven items asking about negative affect: mad, bored, lonely, sad, nervous, worried, and afraid. This Negative Affect factor was supported by confirmatory factor analyses (see S3 Text in **S1 File**). The composite score ranges from 5 to 35, with higher scores representing more negative affect. The alpha coefficient for this scale was .71.

### Perceived stress

To measure stress, we administered the Perceived Stress Scale for Children (PSS-C) [32]. This self-report measure consists of 13 items on a 4-point Likert scale ranging from 'never' to a 'lot.' The questions assess perceived stress related to time pressure, academic performance, and relationships with family and friends. A higher score indicates a greater level of perceived stress. For this study the alpha coefficient was .62.

### Mindfulness

To assess trait mindfulness, we administered the Child and Adolescent Mindfulness Measure (CAMM) [4]. This self-report scale consists of 10 items querying the frequency of non-mindful thoughts or behaviors on a 5-point Likert scale from 'never true' to 'always true.' All items are negatively worded and reverse-scored. Higher scores represent greater levels of mindfulness. Specifically, the authors describe the scale as measuring both awareness of the present and the degree to which one has a nonjudgmental attitude towards one's thoughts and feelings, which includes not suppressing or avoiding them. In this study the alpha coefficient was .77.

### Analysis methods

All analyses were conducted in R programming language. We first calculated descriptive statistics for the scales and demographics. Then we conducted linear multivariate analyses, beginning with the distributions. For scales with more than 3 items, we assumed missing data was at random, and imputed missing item responses (0.17%) with the *mice* package, using predictive mean matching [33]. Then, the data were *z*-scored or *t*-scored where applicable (i.e., the RCADS composites), and outliers were removed if above or below 3 SD of the mean. Resulting distributions were assessed to make sure skewness < |2|. We then assessed bivariate relationships among our main variables of interest. Note that we refer to anxiety, depression, perceived stress and negative affect as mental health outcomes, Child COVID-19 Impact as a predictor, and CAMM as a moderator. We calculated correlations between Child COVID Impact, the four mental health outcomes, and CAMM. We corrected *p*-values across the four outcome measures (i.e., the different mental health symptoms) using false discovery rate correction (FDR) [34].

We then conducted a set of moderation analyses using linear regressions (package: *lm*). Our main hypothesis was that the relationship between children's COVID Impact and their

mental health symptoms would be moderated by mindfulness. We thus estimated four separate linear models, controlling for age, gender, and maternal education. Moderators and predictors were z-transformed and all variables were introduced in one step for estimation. The specific outcome of interest is the interaction between Child COVID Impact and mindfulness. We report fit statistics, slope coefficients (standardized *betas)*, and variance explained. For interpretation of the interaction effects we use median splits for visualization and simple slope analysis [35], which entails probing effects at ±1 SD from the mean of mindfulness.

## Results

### Demographics

The average age of the 163 children (87 male) was 9 years, 5 months (*SD* = 6 months, range 8;2–10;6). The median household income range was 80,000–120,000 US Dollars (see S1 Fig in **S1 File**). Levels of maternal education were high, with a majority having at least a bachelor's degree (see S2 Fig in **S1 File**). The majority of children came from the states of Georgia, Massachusetts, California, and Texas. Parents reported the racial identities of their children—45.4% were White, 17.8% Multiracial, 18.4% Latino/Hispanic, 9.2% Asian, 2.5% Black or African American, 1.2% Other, and 5.5% chose not to respond. By parental report, 10 children had clinical diagnoses of anxiety, 24 were diagnosed with Attention-Deficit/Hyperactivity Disorder (ADHD), and five had been diagnosed with both ADHD and anxiety. By parental report, 69 had previous mindfulness experience (see S3 Fig in **S1 File**).

### Mental health variables

Descriptive statistics for the main variables of interest are shown in **Table 1**, including means and standard deviations of negative affect, RCADS anxiety and depression subscales, PSS-C, CAMM and COVID-Impact. *T*-scores for RCADS were relatively low. On the anxiety scale all children fell in the low severity range (0–65) and on the depression scale only two children fell in the moderate range (65–70), with one falling in the severe range (>70).

### Bivariate relationships

Correlations for the main variables of interest are shown in **Table 1**. Measures of mental health symptoms showed strong intercorrelations (*rs* from .4 to .6, all *ps* < .0001). Mindfulness had strong negative relationships with mental health symptoms (all *ps* < .0001). COVID Impact as

**Table 1. Bivariate correlations and descriptive statistics for main variables of interest.**

| Scale | NAffect | Anxiety | Depression | Stress | CAMM | Mean | S.D. |
|---|---|---|---|---|---|---|---|
| NAffect | | | | | | 14.20 | 4.66 |
| Anxiety | 0.45**** | | | | | 41.32 | 7.17 |
| Depression | 0.42**** | 0.59**** | | | | 43.85 | 8.18 |
| Stress | 0.41**** | 0.49**** | 0.55**** | | | 10.87 | 4.91 |
| CAMM | -0.38**** | -0.50**** | -0.49**** | -0.40**** | | 27.05 | 6.63 |
| COVID-Impact | 0.18* | 0.21* | 0.07 | 0.02 | -0.21* | 14.22 | 4.72 |

NAffect = Negative Affect, Anxiety = RCADS-Anxiety, Depression = RCADS-Depression, Stress = PSSTotal-C, CAMM = Child and Adolescent Mindfulness Measure.

Mean anxiety and depression scores are *t*-scores.

*, *p* < 0.05

**** *p* < 0.0001

**Table 2. Mindfulness is a moderator of the relationship between COVID Impact and negative affect.**

| Model: Negative Affect | β | Standard Error | t value | P | Sig. |
|---|---|---|---|---|---|
| COVID Impact | 0.081 | 0.081 | 0.994 | 0.3222 | |
| CAMM | -0.394 | 0.083 | -4.773 | 0.0000 | *** |
| Gender | -0.396 | 0.160 | -2.482 | 0.0143 | * |
| Age | 0.042 | 0.081 | 0.526 | 0.5996 | |
| Maternal Education | -0.081 | 0.078 | -1.027 | 0.3063 | |
| COVID Impact X CAMM | -0.192 | 0.077 | -2.482 | 0.0143 | * |

Multiple R-squared: 0.2225, Adjusted R-squared: 0.1871.

*, p < 0.05

*** p < 0.001

reported by children showed a smaller but significant positive correlation with negative affect ($r$ (163) = .18, $p$ = .029, FDR-corrected) and RCADS-Anxiety ($r$(163) = .21, $p$ = .028, FDR-corrected), and a similar negative correlation with mindfulness ($r$(163) = -.21, $p$ = .028, FDR-corrected).

### Moderation analyses

Mindfulness significantly moderated the relationship between COVID-Impact and negative affect in the children (**Table 2**). We further investigated this relationship using simple slopes analysis. High mindfulness (+1SD) children showed no relationship between COVID- Impact scores and Negative Affect scores ($\beta$ = -.11, $SE$ = 0.11, $p$ = .31), whereas low mindfulness (-1SD) children had a positive relationship between COVID-Impact scores and Negative Affect scores ($\beta$ = 0.27, $SE$ = 0.11, $p$ = .018). This is also depicted in a median split (**Fig 1**).

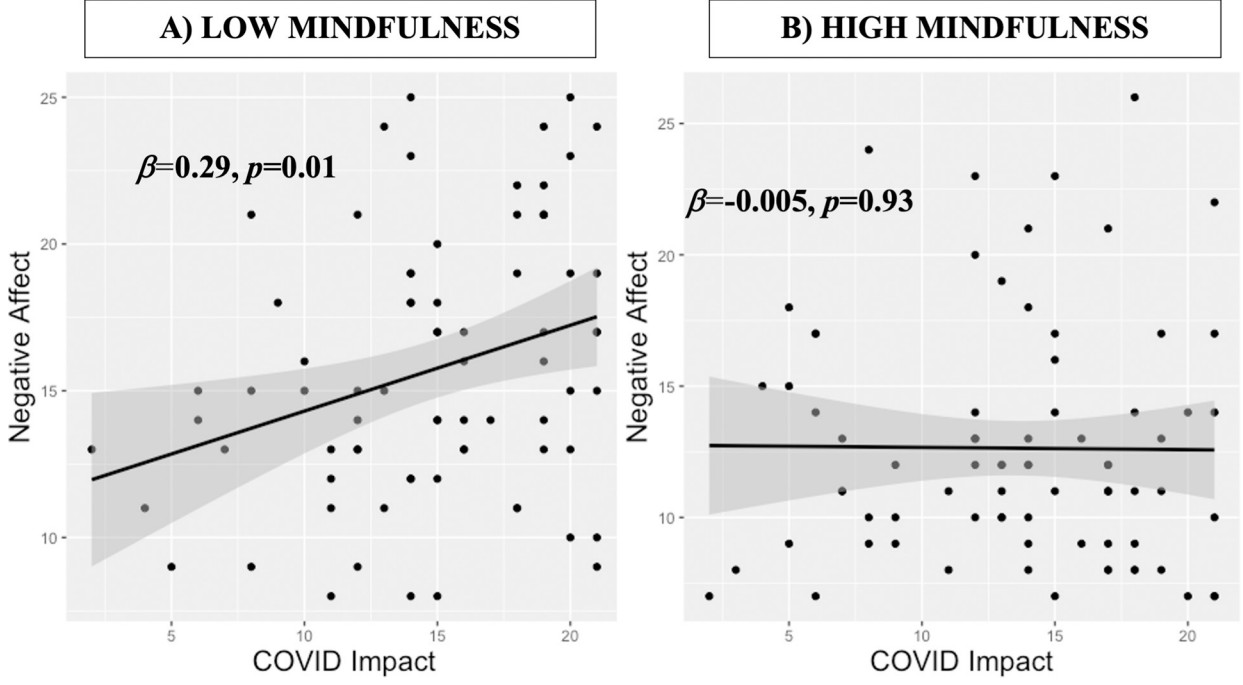

**Fig 1. Relationship between child COVID-Impact and negative affect, median split by mindfulness.** A) Low mindfulness children show positive correlation between COVID Impact and Negative Affect. B) High mindfulness children show no relationship between COVID Impact and Negative Affect. In black are linear regression fits with gray confidence bar, and regression slope and $p$-values are listed in bold.

Gender was the only control variable to reach significance, where female children showed higher negative affect than males. We probed this gender vs. negative affect relationship without the other variables, and it was trend-level in the full sample ($t(161) = -1.8$, $p = .07$). No other main moderation analyses (with depression, anxiety or stress) were significant (see S1-S3 Tables in **S1 File**).

## Discussion

In this study, we found evidence that mindfulness was related to emotional resilience in children. We specifically investigated whether trait mindfulness was associated with more positive mental health outcomes in response to the severe life changes of the COVID-19 pandemic. We obtained self-report data from US children during the height of pandemic distancing restrictions, from July 2020 to February 2021. Greater trait mindfulness was associated with less stress, anxiety, depression, and negative affect. Greater mindfulness was also associated with lower children's ratings on the impact of COVID-19 on their lives. Critically, trait mindfulness moderated the relationship between COVID-19 child impact and negative affect. Children scoring high on mindfulness showed no correlation between rated COVID-19 impact and negative affect, whereas those who scored low on mindfulness showed a positive correlation between child COVID-19 impact and negative affect. Overall, higher levels of trait mindfulness may have helped children to better cope with a wide range of COVID-19 stressors.

To assess COVID-19 impact, we asked children questions like 'how much harder has it been to see friends during the pandemic?" During pandemic-related lockdowns, children may find it harder to see friends and understandably experience negative emotions like frustration and loneliness about such isolation. However, the degree to which they blame themselves for these emotions, try to avoid feeling them, and generally get caught up in their emotions as opposed to having a healthy distance from them, may play a role in overall mental health outcomes. In our study, trait mindfulness was associated with more positive mental health outcomes, as in prior work [4, 5, 25], and lower COVID-19 stress. Perhaps higher trait mindfulness means less self-blame and judgment, which in turn supports emotional resilience to pandemic related stressors.

Two theoretical foundations of mindfulness are attention to the present and non-judgement [20]. Some trait mindfulness scales like the Mindful Awareness Attention Scale (MAAS) focus on the attentional construct [36, 37], whereas the Child and Adolescent Mindfulness Measure (CAMM), administered in this study, focuses on the construct of non-judgement. The CAMM aims to capture propensities to self-blame and judge one's feelings (e.g., "I get upset with myself for having certain thoughts"). Lower scores on the CAMM correlate with less adaptive strategies to cope with one's emotions, such as rumination, catastrophizing, and suppression [4, 25]. Consistent with these prior studies, our study showed that mindfulness moderated the relationship between pandemic stress and negative affect. Mindfulness may support emotional resilience through the mechanisms of non-judgement and acceptance of emotions.

This study connects mindfulness to positive mental health outcomes in children during the pandemic. This finding is consistent with studies showing that self-reported mindfulness correlated with lower psychological distress in adults during the pandemic [38–41], as well as an RCT study showing that a four-week mobile-phone mindfulness intervention reduced anxiety symptoms in university students under quarantine [42]. Mindfulness interventions may help children cope with adverse events like the COVID-19 pandemic.

Several limitations of this study may be noted. First, the children and their families were self-selected participants in a multi-week intervention study. Self-selection yielded a more

educated and higher income sample than originally intended. Most of the children were from high SES families, and thus may not represent the kinds of children who were most vulnerable to the life disruptions of the COVID-19 pandemic. In fact, on average, children in this study reported lower rates of clinical anxiety and depression than norms for the measures (mean *T*-scores for our sample were 42% and 43% respectively). Some studies have reported that the sudden and severe changes on account of the pandemic, such as school closures and social isolation from peers, had broad, negative influences on children [43–45]. The children in the present study, however, appeared to be adapting well to the pandemic (although we cannot exclude the possibility of children wanting to report socially desirable responses). Other limitations were that the COVID-specific questionnaire was invented for the study and not standardized, and that remote test administration is not standard for the field.

A final limitation is that this is a correlational study that cannot determine that mindfulness causes emotional resilience. Moderation analyses cannot provide cause-and-effect evidence (rather, they show how a relationship between survey constructs can be altered by a third variable). Other studies on trait mindfulness and resilience have adopted different approaches. For example, Park et al. assessed COVID-related distress in adults at three time-points [46]. They found baseline mindfulness not only correlated with less distress at follow-up, but also correlated with more substantial decreases in distress over time. Similar longitudinal studies in children could be used to best assess whether mindfulness promotes emotional resilience *in response* to COVID-19 stressors.

Despite these limitations, the present finding that trait mindfulness moderated the relationship between pandemic stress and negative affect suggests that mindfulness supports resilience in the face of adverse experience. There is substantial evidence in adults that trait mindfulness promotes resilience to COVID-19 stressors [46–51]. For example, Vos et al. found that worry about COVID-19 was only associated with anxiety for adults with low but not high mindfulness [50]. What is the specific mechanism by which mindfulness promotes emotional resilience in the face of adversity? One idea is that mindfully treating experiences in a non-judgmental and accepting manner could help individuals decrease depressive rumination [23, 24]. Alternatively, or additionally, mindfulness practices like body scans that involve present-moment attention to body sensations may be key to the benefits of mindfulness. These practices may support more accurate appraisals of bodily sensations and diminish anxiety [22, 52, 53]. Further mindfulness research could help identify which cognitive and affective traits of children allow them to meaningfully and constructively respond to emotional challenges.

The finding that mindfulness in children moderated negative affect in response to the sudden, comprehensive, and chronic disruptions of school and home life brought on by a global pandemic, further motivates research into the implementation of mindfulness instruction for children. The present findings are consistent with evidence from randomized controlled trials that mindfulness interventions can reduce stress and negative affect in children [6]. In one study, reductions in stress after a school-based mindfulness intervention were associated with changes in brain function [54]. Taken together, these findings encourage the adoption of mindfulness instruction in schools or other settings to help children be more emotionally resilient. More resources for promoting mental health in children are provided by the American Academy of Child and Adolescent Psychiatry (https://rb.gy/i0ba4).

## Conclusions

We conducted a cross-sectional study with 163 eight-to-ten-year-old children during the height of the COVID-19 Pandemic. We assessed relationships between child-reported mindfulness and several measures of mental health outcomes like stress, anxiety, and negative affect,

along with a specifically designed measure of COVID-19 impact on their lives. Ultimately, we found mindfulness was associated with resilience, defined as positive mental health outcomes in the face of adversity. These findings encourage the adoption of mindfulness instruction in schools or other settings to help children be more emotionally resilient.

## Supporting information

**S1 File. Contains all text, figures, and tables.**
(DOCX)

## Acknowledgments

We thank our testers: Amanda Miller, David Bates, Ross Weissman, Joohee Baik, June Okada, William Oliver, and Harriet Richards. We also thank Xochitl Arechiga, Yesi Camacho Torres, Hope Kentala, Natalie Gardino, Jeff Dieffenbach, for helping to run the study, Yilei Chen for data preprocessing, and Sadie Zacharek for helpful comments on the manuscript. We thank the Hock E. Tan and K. Lisa Yang Center for Autism Research at Massachusetts Institute of Technology.

## Author Contributions

**Conceptualization:** Isaac N. Treves, John D. E. Gabrieli.

**Formal analysis:** Isaac N. Treves.

**Funding acquisition:** Ola Ozernov-Palchik, Halie A. Olson.

**Investigation:** Isaac N. Treves.

**Methodology:** Isaac N. Treves.

**Project administration:** Cindy E. Li, Kimberly L. Wang, Ola Ozernov-Palchik, Halie A. Olson.

**Supervision:** Ola Ozernov-Palchik, Halie A. Olson, John D. E. Gabrieli.

**Writing – original draft:** Isaac N. Treves, Cindy E. Li, John D. E. Gabrieli.

**Writing – review & editing:** Isaac N. Treves, Cindy E. Li, Kimberly L. Wang, Ola Ozernov-Palchik, Halie A. Olson, John D. E. Gabrieli.

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
