## [Decision Letter · Decision Letter 0]

25 Apr 2023

PONE-D-22-31625Mindfulness supports emotional resilience in children during the COVID-19 PandemicPLOS ONE

Dear Dr. Treves,

Thank you for submitting your manuscript to PLOS ONE. After careful consideration, we feel that it has merit but does not fully meet PLOS ONE’s publication criteria as it currently stands. Therefore, we invite you to submit a revised version of the manuscript that addresses the points raised during the review process.

We look forward to receiving your revised manuscript.

Kind regards,

Lucinda Shen, MSc

Staff Editor

PLOS ONE

Journal Requirements:

5. Please ensure that you refer to Figure 1 in your text as, if accepted, production will need this reference to link the reader to the figure.

6. Please upload a copy of Figure 2, to which you refer in your text on page 18. If the figure is no longer to be included as part of the submission please remove all reference to it within the text.

Additional Editor Comments:

The mansucript has been evaluated by 6 external reviewers and their comments are seen below. The reviewers have congratulated the authors on an excellent mansucript. They feel that the results of the work further scientific understanding in the field.

One reviewer has raised a minor query about the regarding the demographic of the study participants and suggested a discussion on whether schools provided mindfulness training to students

Please could you address this comment.

Reviewers' comments:

Reviewer's Responses to Questions

**Comments to the Author**

1. Is the manuscript technically sound, and do the data support the conclusions?

Reviewer #1: Yes

Reviewer #2: Yes

Reviewer #3: Yes

Reviewer #4: Yes

Reviewer #5: Yes

Reviewer #6: Yes

2. Has the statistical analysis been performed appropriately and rigorously? 

Reviewer #1: Yes

Reviewer #2: Yes

Reviewer #3: Yes

Reviewer #4: Yes

Reviewer #5: Yes

Reviewer #6: Yes

3. Have the authors made all data underlying the findings in their manuscript fully available?

Reviewer #1: Yes

Reviewer #2: Yes

Reviewer #3: Yes

Reviewer #4: Yes

Reviewer #5: Yes

Reviewer #6: Yes

4. Is the manuscript presented in an intelligible fashion and written in standard English?

Reviewer #1: Yes

Reviewer #2: Yes

Reviewer #3: Yes

Reviewer #4: Yes

Reviewer #5: Yes

Reviewer #6: Yes

5. Review Comments to the Author

Reviewer #1: Children's lives were dramatically and abruptly changed by the COVID-19 pandemic. Stress, anxiety, depression, and other mental health issues arose as a direct result of these shifts. Increased awareness of one's emotional bodily sensations and a less critical or judgemental outlook on one's experiences are both benefits of practicing mindfulness. This research shows that children who practice mindfulness during a pandemic have better mental health as a result. In line with previous research, this data indicates that those who reported higher levels of mindfulness experienced less emotional discomfort during the pandemic.

The children and their families volunteered to take part in the multi-week intervention research, which is one of the drawbacks the authors have discussed. By choosing themselves, the participants in this study have higher levels of education and money than was anticipated, which could have skewed the findings in favor of the research question. The authors correctly point out that they cannot rule out the potential that the findings of this study would be influenced by the fact that the participants were children who were trying to report socially desirable responses. This is only a correlational study, therefore it cannot prove that mindfulness is what actually makes people more resilient to negative emotions.

Furthermore, moderation analyses do not produce proof of causation.

Despite these caveats, the findings are still pertinent, and it is hoped that publication of the findings may encourage doctors and educators to employ Mindfulness approaches when working with children.

Reviewer #2: Authors have done a cross-sectional study to investigate whether trait mindfulness was related to emotional resilience in response to stressful changes in education and home-life during the COVID-19 pandemic in the United States. They have used surrogate markers toidentify resilience.

Introduction is well written, highlights the need and importance of current study and providing some background data. It is an important topic and authors have accordingly highlighted it.

Methods are well structuresd, clearly explaining the data collection process, scales used and their questionnaire and its interpretation. Statistical analyses is wisely used to identify potential relationship, after normalizing the data and removing the outliers that can often drive false results in these kind of analyses.

Results are clearly explained. Tables and figures are very useful to see the relationships. Their results do highlight the important corelation that authors hypothesized.

Discussion and conlusion is robust, they have done a great job in explaining what their results signifies and previous studies on similar topics. It was also interesting to note the gender differences that were observed in the current stuies. Limitations of the study are accurately identified and mentioned.

I would like to congratulate the authors on conducting this important study, meticulously and accurately reporting their findings with their limitations.

Reviewer #3: Thank you for submitting this manuscript for review. It was quite well written and comprehensive. The results help confirm association of mindfulness with resilience and serves an important purpose in understanding the importance of inclusion of mindfulness in school curricula. The virtual nature of the study is also interesting though you rightly pointed out some of the limitations with it.

Reviewer #4: This is a well written article with a clear research question. The study performs a good analysis of the data it presents. The study utilized a good method, and the study design appears clear and easy to replicate. There does not appear to be any bias or error. The study identifies its limitations well, and the discussion section is well written.

This article suffers from lacking a conclusion section. Having a concise conclusion section to identify pertinent findings of this study will significantly increase its relevance to the average clinician.

Minor concern

1. line 183 mentions "All analyses were conducted in R." Reviewer was unsure of the meaning of this sentence

Reviewer #5: Congratulations to the authors for their rigorous work on this vital subject matter. If any participants had a psychiatric diagnosis or received any mental health treatment. It would be interesting to know if the school provided mindfulness training to students. The author could comment on the same.

Reviewer #6: Thank you for shedding light on this vital topic. I have read this study's relationship between mindfulness and mental health resilience with great interest. The study population has lower rates of clinical anxiety and depression than norms, so any result extrapolation to draw a general conclusion should be considered with caution. However, the authors adequately explained the limitations.

The introduction is well-framed, and the methodology discusses the covid-specific questionnaire and data-gathering process in terms of clinical presentation, known etiology, and neurobiology at length. Paper may benefit from adding any specific resources, if they exist (CDC, AACAP, etc.), that clinicians can access to learn more about it or seek guidance.

6. PLOS authors have the option to publish the peer review history of their article (what does this mean?). If published, this will include your full peer review and any attached files.

Reviewer #1: No

Reviewer #2: No

Reviewer #3: No

Reviewer #4: **Yes: **Lakshit Jain MD

Reviewer #5: No

Reviewer #6: No

---

## [Author Response · Author response to Decision Letter 0]

10 May 2023

Response summary: 

We thank the editor and reviewers for their useful comments. We have included a concise conclusion, provided more information about children’s previous mindfulness experience and demographics, and included a link for mental health resources. Additionally, we have updated the formatting of the manuscript and supplement to match PLOSOne’s style guide.

Reviewer comments

Reviewer #1: Children's lives were dramatically and abruptly changed by the COVID-19 pandemic. Stress, anxiety, depression, and other mental health issues arose as a direct result of these shifts. Increased awareness of one's emotional bodily sensations and a less critical or judgemental outlook on one's experiences are both benefits of practicing mindfulness. This research shows that children who practice mindfulness during a pandemic have better mental health as a result. In line with previous research, this data indicates that those who reported higher levels of mindfulness experienced less emotional discomfort during the pandemic.

The children and their families volunteered to take part in the multi-week intervention research, which is one of the drawbacks the authors have discussed. By choosing themselves, the participants in this study have higher levels of education and money than was anticipated, which could have skewed the findings in favor of the research question. The authors correctly point out that they cannot rule out the potential that the findings of this study would be influenced by the fact that the participants were children who were trying to report socially desirable responses. This is only a correlational study, therefore it cannot prove that mindfulness is what actually makes people more resilient to negative emotions.

Furthermore, moderation analyses do not produce proof of causation.

Despite these caveats, the findings are still pertinent, and it is hoped that publication of the findings may encourage doctors and educators to employ Mindfulness approaches when working with children.

Response:

Thank you for your comments and your summary of the relevance of the paper. 

Reviewer #2: Authors have done a cross-sectional study to investigate whether trait mindfulness was related to emotional resilience in response to stressful changes in education and home-life during the COVID-19 pandemic in the United States. They have used surrogate markers to identify resilience.

Introduction is well written, highlights the need and importance of current study and providing some background data. It is an important topic and authors have accordingly highlighted it.

Methods are well structuresd, clearly explaining the data collection process, scales used and their questionnaire and its interpretation. Statistical analyses is wisely used to identify potential relationship, after normalizing the data and removing the outliers that can often drive false results in these kind of analyses.

Results are clearly explained. Tables and figures are very useful to see the relationships. Their results do highlight the important corelation that authors hypothesized.

Discussion and conlusion is robust, they have done a great job in explaining what their results signifies and previous studies on similar topics. It was also interesting to note the gender differences that were observed in the current stuies. Limitations of the study are accurately identified and mentioned.

I would like to congratulate the authors on conducting this important study, meticulously and accurately reporting their findings with their limitations.

Response: 

Thank you for your comments. 

Reviewer #3: Thank you for submitting this manuscript for review. It was quite well written and comprehensive. The results help confirm association of mindfulness with resilience and serves an important purpose in understanding the importance of inclusion of mindfulness in school curricula. The virtual nature of the study is also interesting though you rightly pointed out some of the limitations with it.

Response: 

Thank you for your comments. 

Reviewer #4: This is a well written article with a clear research question. The study performs a good analysis of the data it presents. The study utilized a good method, and the study design appears clear and easy to replicate. There does not appear to be any bias or error. The study identifies its limitations well, and the discussion section is well written.

This article suffers from lacking a conclusion section. Having a concise conclusion section to identify pertinent findings of this study will significantly increase its relevance to the average clinician.

Minor concern

1. line 183 mentions "All analyses were conducted in R." Reviewer was unsure of the meaning of this sentence

Response:

Thank you for your comments and notes. R is a programming language for statistical computing. For clarity, we have added “All analyses were conducted in R programming language. “

We have also added a concise conclusion section: 

“We conducted a cross-sectional study with 163 eight-to-ten-year-old children during the height of the COVID-19 Pandemic. We assessed relationships between child-reported mindfulness and several measures of mental health outcomes like stress, anxiety, and negative affect, along with a specifically designed measure of COVID-19 impact on their lives. Ultimately, we found mindfulness was associated with resilience, defined as positive mental health outcomes in the face of adversity. These findings encourage the adoption of mindfulness instruction in schools or other settings to help children be more emotionally resilient. “ 

Reviewer #5: Congratulations to the authors for their rigorous work on this vital subject matter. If any participants had a psychiatric diagnosis or received any mental health treatment. It would be interesting to know if the school provided mindfulness training to students. The author could comment on the same.

Response: Thank you for your comments and questions. We asked caregivers whether the children previously received any mindfulness training at home, school, or somewhere else. If they answered yes, they were asked several follow-up questions about how many sessions they partook in training, what the training generally consisted of, and whether they continued to practice mindfulness (or SEL) on their own or with anyone else, and if so how often. We summarized the total number of children with previous mindfulness experience, at line 268. In addition, we have included a flowchart with a breakdown of previous mindfulness experience by diagnostic category, in the revised supplement, figure 3. 

Reviewer #6: Thank you for shedding light on this vital topic. I have read this study's relationship between mindfulness and mental health resilience with great interest. The study population has lower rates of clinical anxiety and depression than norms, so any result extrapolation to draw a general conclusion should be considered with caution. However, the authors adequately explained the limitations.

The introduction is well-framed, and the methodology discusses the covid-specific questionnaire and data-gathering process in terms of clinical presentation, known etiology, and neurobiology at length. Paper may benefit from adding any specific resources, if they exist (CDC, AACAP, etc.), that clinicians can access to learn more about it or seek guidance.

Response: Thank you for your comments. We have inserted a link to mental health resources from the American Academy of Child and Adolescent Psychiatry in the discussion section. https://www.aacap.org/AACAP/Families_Youth/Resource_Libraries/Coronavirus_Resource_Library/AACAP/Families_and_Youth/Resource_Libraries/Coronavirus.aspx?hkey=e254d570-c7cf-412b-833e-bb342ac4f312

---

## [Editor Report · Decision Letter 1]

12 Jun 2023

Mindfulness supports emotional resilience in children during the COVID-19 Pandemic

PONE-D-22-31625R1

Dear Dr. Treves,

We’re pleased to inform you that your manuscript has been judged scientifically suitable for publication and will be formally accepted for publication once it meets all outstanding technical requirements.

Kind regards,

Dario Ummarino, PhD

Senior Editor

PLOS ONE

---

## [Editor Report · Acceptance letter]

4 Jul 2023

PONE-D-22-31625R1 

Mindfulness supports emotional resilience in children during the COVID-19 Pandemic 

Dear Dr. Treves:

I'm pleased to inform you that your manuscript has been deemed suitable for publication in PLOS ONE. Congratulations! Your manuscript is now with our production department. 

Kind regards, 

on behalf of

Dr Dario Ummarino, PhD 

Staff Editor

PLOS ONE